# One-Shot Neural Cross-Lingual Transfer for Paradigm Completion

## Abstract

We present a novel cross-lingual transfer method for paradigm completion, the task of mapping a lemma to its inflected forms, using a neural encoder-decoder model, the state of the art for the monolingual task. We use labeled data from a high-resource language to increase performance on a low-resource language. In experiments on 21 language pairs from four different language families, we obtain up to 58% higher accuracy than without transfer and show that even zero-shot and one-shot learning are possible. We further find that the degree of language relatedness strongly influences the ability to transfer morphological knowledge.

## 1 Introduction

Low-resource natural language processing remains an open problem for many tasks of interest. Furthermore, for most languages in the world, high-cost linguistic annotation and resource creation are unlikely to be undertaken in the near future. In the case of morphology, out of the 7000 currently spoken (Lewis, 2009) languages, only about 200 have computer-readable annotations (Sylak-Glassman et al., 2015) – although morphology is easy to annotate compared to syntax and semantics. *Transfer learning* is one solution to this problem: it exploits annotations in a high-resource language to train a system for a low-resource language. In this work, we present a method for cross-lingual transfer of inflectional morphology using an encoder-decoder recurrent neural network (RNN). This allows for the development of tools for computational morphology with limited annotated data.

In morphologically rich languages, individual lexical entries may be realized as distinct inflec-

|   | PresInd |  | PastInd |  |
|---|---------|---------|---------|---------|
|   | Sg | Pl | Sg | Pl |
| 1 | *sueño* | *soñamos* | *soñé* | *soñamos* |
| 2 | *sueñas* | *soñáis* | *soñaste* | *soñasteis* |
| 3 | *sueña* | *sueñan* | *soñó* | *soñaron* |

Table 1: Partial inflection table for the Spanish verb *soñar*

tions of a single lemma depending on the syntactic context. For example, the 3SgPresInd of the English verbal lemma *to bring* is *brings*. In many languages, a lemma can have hundreds of individual forms. Thus, both generation and analysis of such morphological inflections are active areas of research in NLP and morphological processing has been shown to be a boon to several other down-stream applications, e.g., machine translation (Dyer et al., 2008), speech recognition (Creutz et al., 2007), parsing (Seeker and Çetinoğlu, 2015), keyword spotting (Narasimhan et al., 2014) and word embeddings (Cotterell et al., 2016b), *inter alia*. In this work we focus on paradigm completion, a form of morphological generation that maps a given lemma to a target inflection, e.g., (*bring*, Past) $\mapsto$ *brought* (with Past being the target tag).

RNN sequence-to-sequence models (Sutskever et al., 2014; Bahdanau et al., 2015) are the state of the art for paradigm completion (Faruqui et al., 2016; Kann and Schütze, 2016a; Cotterell et al., 2016a). However, these models require a large amount of data to achieve competitive performance; this makes them unsuitable for out-of-the-box application to paradigm completion in the low-resource scenario. To mitigate this, we consider transfer learning: we train an end-to-end neural system jointly with limited data from a low-resource language and a larger amount of data from a high-resource language. This technique allows the model to apply knowledge distilled from the high-resource training data to the low-resource language as needed.

We conduct experiments on 21 language pairs from four language families, emulating a low-resource setting. Our results demonstrate successful transfer of morphological knowledge. We show improvements in accuracy and edit distance of up to 58% (accuracy) and 4.62 (edit distance) over the same model with only in-domain language data on the paradigm completion task. We further obtain up to 44% (resp. 14%) improvement in accuracy for the one-shot (resp. zero-shot) setting, i.e., one (resp. zero) in-domain language sample per target tag. We also show that the effectiveness of morphological transfer depends on language relatedness, measured by lexical similarity.

## 2 Inflectional Morphology and Paradigm Completion

Many languages exhibit inflectional morphology, i.e., the form of an individual lexical entry mutates to show properties such as person, number or case. The citation form of a lexical entry is referred to as the **lemma** and the collection of its possible inflections as its **paradigm**. Tab. 1 shows an example of a partial paradigm; we display several forms for the Spanish verbal lemma *soñar*. We may index the entries of a paradigm by a **morphological tag**, e.g., the 2SgPresInd form *sueñas* in Tab. 1. In generation, the speaker must select an entry of the paradigm given the form's context. In general, the presence of rich inflectional morphology is problematic for NLP systems as it greatly increases the token-type ratio and, thus, word form sparsity.

An important task in inflectional morphology is **paradigm completion** (Durrett and DeNero, 2013; Ahlberg et al., 2014; Nicolai et al., 2015; Cotterell et al., 2015; Faruqui et al., 2016). Its goal is to map a lemma to all individual inflections, e.g., (*soñar*, 1SgPresInd) $\mapsto$ *sueño*. There are good solutions for paradigm completion when a large amount of annotated training data is available (Cotterell et al., 2016a).[1] In this work, we address the low-resource setting, an up to now unsolved challenge.

### 2.1 Transferring Inflectional Morphology

In comparison to other NLP annotations, e.g., part-of-speech (POS) and named entities, morphological inflection does not lend itself easily to transfer.

---

[1] The SIGMORPHON 2016 shared task (Cotterell et al., 2016a) on morphological reinflection, a harder generalization of paradigm completion, found that $\geq 98\%$ accuracy can be achieved in many languages with neural sequence-to-sequence models, improving the state of the art by $10\%$.

We can define a universal set of POS tags (Petrov et al., 2012) or of entity types (e.g., coarse-grained types like *person* and *location* or fine-grained types (Yaghoobzadeh and Schütze, 2015)), but inflection is much more language-specific. It is infeasible to transfer morphological knowledge from Chinese to Portuguese as Chinese does not use inflected word forms. Transferring named entity recognition, however, among Chinese and European languages works well (Wang and Manning, 2014a). But even transferring inflectional paradigms from morphologically rich Arabic to Portuguese seems difficult as the inflections often mark dissimilar subcategories. In contrast, transferring morphological knowledge from Spanish to Portuguese, two languages with similar conjugations and 89% lexical similarity, appears promising. Thus, we conjecture that transfer of inflectional morphology is only viable among *related languages*.

### 2.2 Formalization of the Task

We now offer a formal treatment of the cross-lingual paradigm completion task and develop our notation. Let $\Sigma_\ell$ be a discrete alphabet for language $\ell$ and let $\mathcal{T}_\ell$ be a set of morphological tags for $\ell$. Given a lemma $w_\ell$ in $\ell$, the morphological paradigm (inflectional table) $\pi$ can be formalized as a set of pairs

$$\pi(w_\ell) = \left\{ \left( f_k[w_\ell], t_k \right) \right\}_{k \in T(w_\ell)} \quad (1)$$

where $f_k[w_\ell] \in \Sigma_\ell^+$ is an inflected form, $t_k \in \mathcal{T}_\ell$ is its morphological tag and $T(w_\ell)$ is the set of slots in the paradigm; e.g., a Spanish paradigm is:

$$\pi\left(\text{SOÑAR}\right) = \left\{ \left( \textit{sueño}, \text{1SgPresInd} \right), \dots, \left( \textit{soñaran}, \text{3PlPastSbj} \right) \right\}$$

Paradigm completion consists of predicting the entire paradigm $\pi(w_\ell)$ given the lemma $w_\ell$.

In cross-lingual paradigm completion, we consider a *high-resource source language* $\ell_s$ (lots of training data available) and a *low-resource target language* $\ell_t$ (little training data available). We denote the source training examples as $\mathcal{D}_s$ (with $|\mathcal{D}_s| = n_s$) and the target training examples as $\mathcal{D}_t$ (with $|\mathcal{D}_t| = n_t$). The goal of cross-lingual paradigm completion is to populate paradigms in the low-resource target language with the help of data from the high-resource source language, using only few in-domain examples.

## 3 Cross-Lingual Transfer as Multi-Task Learning

We describe our probability model for morphological transfer using terminology from multi-task learning (Caruana, 1997; Collobert et al., 2011). We consider two tasks, training a paradigm completor (i) for a high-resource language and (ii) for a low-resource language. We want to train jointly so we reap the benefits of having related languages. Thus, we define the log-likelihood as

$$\mathcal{L}(\boldsymbol{\theta}) = \sum_{(k, w_{\ell_t}) \in \mathcal{D}_t} \log p_{\boldsymbol{\theta}}\left(f_k[w_{\ell_t}] \mid w_{\ell_t}, t_k\right) \quad (2)$$
$$+ \sum_{(k, w_{\ell_s}) \in \mathcal{D}_s} \log p_{\boldsymbol{\theta}}(f_k[w_{\ell_s}] \mid w_{\ell_s}, t_k)$$

where we tie parameters $\boldsymbol{\theta}$ for the two languages together to allow the transfer of morphological knowledge between languages. Each probability distribution $p_{\boldsymbol{\theta}}$ defines a distribution over all possible realizations of an inflected form, i.e., a distribution over $\Sigma^*$. For example, consider the related Romance languages Spanish and French; focusing on one term from each of the summands in Eq. (2) (the past participle of the translation of *to visit* in each language), we arrive at

$$\mathcal{L}_{\text{visit}}(\boldsymbol{\theta}) = \log p_{\boldsymbol{\theta}}(\textit{visitado} \mid \textit{visitar}, \mathsf{PastPart})$$
$$+ \log p_{\boldsymbol{\theta}}(\textit{visité} \mid \textit{visiter}, \mathsf{PastPart}) \quad (3)$$

Our *cross-lingual* setting forces both transductions to share part of the parameter vector $\boldsymbol{\theta}$, to represent morphological regularities between the two languages in a common embedding space and, thus, to enable morphological transfer. This is no different from *monolingual* multi-task settings, e.g., jointly training a parser and tagger for transfer of syntax.

Based on recent advances in neural transducers, we parameterize each distribution as an encoder-decoder RNN, as in (Kann and Schütze, 2016b). In their setup, the RNN encodes the input and predicts the forms in a *single* language. In contrast, we force the network to predict *two* languages.

### 3.1 Encoder-Decoder RNN

We parameterize the distribution $p_{\boldsymbol{\theta}}$ as an encoder-decoder gated RNN with attention (Bahdanau et al., 2015), the state-of-the-art solution for the monolingual case (Kann and Schütze, 2016b). A bidirectional gated RNN encodes the input sequence (Cho et al., 2014) – the concatenation of (i) the language

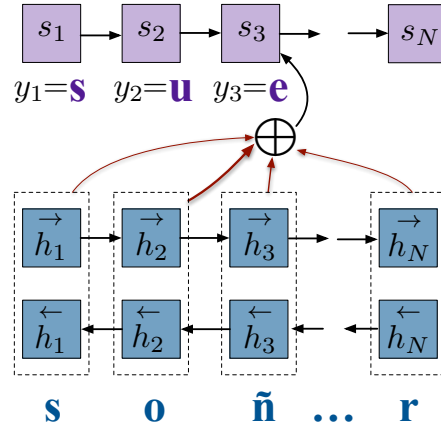

Figure 1: Encoder-decoder RNN for paradigm completion. The lemma *soñar* is mapped to a target form (e.g., *sueña*). For brevity, language and target tags are omitted from the input. Thickness of red arrows symbolizes the degree to which the model attends to the corresponding hidden state of the encoder.

tag, (ii) the morphological tag of the form to be generated and (iii) the characters of the input word – represented by embeddings. The input to the decoder consists of concatenations of $\overrightarrow{h_i}$ and $\overleftarrow{h_i}$, the forward and backward hidden states of the encoder. The decoder, a unidirectional RNN, uses attention: it computes a weight for each $h_i$. Each weight reflects the importance given to that input position. Using the attention weights $\alpha_{ij}$, the probability of the output sequence given the input sequence is:

$$p(y \mid x_1, \ldots, x_{|X|}) = \prod_{t=1}^{|Y|} g(y_{t-1}, s_t, c_t) \quad (4)$$

where $y = (y_1, \ldots, y_{|Y|})$ is the output sequence (a sequence of $|Y|$ characters), $x = (x_1, \ldots x_{|X|})$ is the input sequence (a sequence of $|X|$ characters), $g$ is a non-linear function, $s_t$ is the hidden state of the decoder and $c_t$ is the sum of the encoder states $h_i$, weighted by attention weights $\alpha_i(s_{t-1})$ which depend on the decoder state:

$$c_t = \sum_{i=1}^{|X|} \alpha_i(s_{t-1}) h_i \quad (5)$$

Fig. 1 shows the encoder-decoder. See Bahdanau et al. (2015) for further details.

### 3.2 Input Format

Each source form is represented as a sequence of characters; each character is represented as an embedding. In the same way, each source tag is represented as a sequence of subtags, and each subtag is represented as an embedding. More formally,

we define the alphabet $\Sigma = \cup_{\ell \in L} \Sigma_\ell$ as the set of characters in the languages in $L$, with $L$ being the set of languages in the given experiment. Next, we define $\mathcal{S}$ as the set of subtags that occur as part of the set of morphological tags $\mathcal{T} = \cup_{\ell \in L} \mathcal{T}_\ell$; e.g., if 1SgPresInd $\in \mathcal{T}$, then 1, Sg, Pres, Ind $\in \mathcal{S}$. Note that the set of subtags $\mathcal{S}$ is defined as attributes from the UNIMORPH schema (Sylak-Glassman, 2016) and, thus, is universal across languages; the schema is derived from research in linguistic typology.[2] The format of the input to our system is $\mathcal{S}^+\Sigma^+$. The output format is $\Sigma^+$. Both input and output are padded with distinguished BOW and EOW symbols.

What we have described is the representation of Kann and Schütze (2016b). In addition, we preprend a symbol $\lambda \in L$ to the input string (e.g., $\lambda = $ Es, also represented by an embedding), so the RNN can handle multiple languages simultaneously and generalize over them.

## 4 Languages and Language Families

To verify the applicability of our method to a wide range of languages, we perform experiments on example languages from several different families.

**Romance languages**, a subfamily of Indo-European, are widely spoken, e.g., in Europe and Latin America. Derived from the common ancestor Vulgar Latin (Harris and Vincent, 2003), they share large parts of their lexicon and inflectional morphology; we expect knowledge among them to be easily transferable.

We experiment on Catalan, French, Italian, Portuguese and Spanish. Tab. 2 shows that Spanish – which takes the role of the low-resource language in our experiments – is closely related with the other four, with Portuguese being most similar. We hypothesize that the transferability of morphological knowledge between source and target corresponds to the degree of lexical similarity; thus, we expect Portuguese and Catalan to be more beneficial for Spanish than Italian and French.

The Indo-European **Slavic language family** has its origin in eastern-central Europe (Corbett and Comrie, 2003). We experiment on Bulgarian, Macedonian, Russian and Ukrainian (Cyrillic script) and on Czech, Polish and Slovene (Latin script). Macedonian and Ukranian are low-resource

---

[2]Note that while the subtag set is universal, *which* subtags a language actually uses is language-specific; e.g., Spanish does not mark animacy as Russian does. We contrast this with the universal POS set (Petrov et al., 2012), where it is reasonable to expect that we see all 17 tags in every language.

| | PT | CA | IT | FR |
|---|---|---|---|---|
| similarity to ES | 89% | 85% | 82% | 75% |

Table 2: Lexical similarities for Romance (Lewis, 2009)

languages, so we assign them the low-resource role. For Romance and for Uralic, we experiment with groups containing three or four source languages. To arrive at a comparable experimental setup for Slavic, we run two experiments, each with three source and one target language: (i) from Russian, Bulgarian and Czech to Macedonian; and (ii) from Russian, Polish and Slovene to Ukrainian.

We hope that the paradigm completor learns similar embeddings for, say, the characters "e" in Polish and "$\epsilon$" in Ukrainian. Thus, the use of two scripts in Slavic allows us to explore transfer across different alphabets.

We further consider a non-Indo-European language family, the **Uralic languages**. We experiment on the three most commonly spoken languages – Finnish, Estonian and Hungarian (Abondolo, 2015) – as well as Northern Sami, a language used in Northern Scandinavia. While Finnish and Estonian are closely related (both are members of the Finnic subfamily), Hungarian is a more distant cousin. Estonian and Northern Sami are low-resource languages, so we assign them the low-resource role, resulting in two groups of experiments: (i) Finnish, Hungarian and Estonian to Northern Sami; (ii) Finnish, Hungarian and Northern Sami to Estonian.

**Arabic (baseline)** is a Semitic language (part of the Afro-Asiatic family (Hetzron, 2013)) that is spoken in North Africa, the Arabian Peninsula and other parts of the Middle East. It is unrelated to all other languages used in this work. Both in terms of form (new words are mainly built using a templatic system) and categories (it has tags such as construct state), Arabic is very different. Thus, we do not expect it to support morphological knowledge transfer and we use it as a baseline for all target languages.

## 5 Experiments

We run three experiments on 21 distinct pairings of languages to show the feasibility of morphological transfer and analyze our method. We first discuss details common to all experiments.

We keep **hyperparameters** during all experiments (and for all languages) fixed to the following values. Encoder and decoder RNNs each have 100

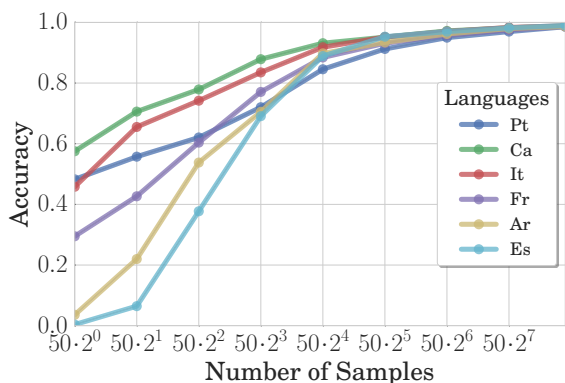

Figure 2: Learning curves showing the accuracy on Spanish test when training on language $\lambda \in \{$Pt, Ca, It, Fr, Ar, Es$\}$. Except for $\lambda=$Es, each model is trained on 12,000 samples from $\lambda$ and "Number of Samples" (x-axis) of Spanish.

hidden units and the size of all subtag, character and language embeddings is 300. For training we use ADADELTA (Zeiler, 2012) with minibatch size 20. All models are trained for 300 epochs. Following Le et al. (2015), we initialize all weights in the encoder, decoder and the embeddings except for the GRU weights in the decoder to the identity matrix. Biases are initialized to zero.

**Evaluation metrics:** (i) 1-best accuracy: the percentage of predictions that match the true answer exactly; (ii) average edit distance between prediction and true answer. The two metrics differ in that accuracy gives no partial credit and incorrect answers may be drastically different from the annotated form without incurring additional penalty. In contrast, edit distance gives partial credit for forms that are closer to the true answer.

## 5.1 Exp. 1: Transfer Learning for Paradigm Completion

In this experiment, we investigate to what extent our model transfers morphological knowledge from a high-resource source language to a low-resource target language. We experimentally answer three questions. (i) Is transfer learning possible for morphology? (ii) How much annotated data do we need in the low-resource target language? (iii) How closely related must the two languages be to achieve good results?

**Data.** Based on complete inflection tables from unimorph.org (Kirov et al., 2016), we create datasets as follows. Each training set consists of 12,000 samples in the high-resource source language and $n_t \in \{50, 200\}$ samples in the low-resource target language. We create target lan-

guage dev and test sets of sizes 1600 and 10,000, respectively.[3] For Romance and Arabic, we create learning curves for $n_t \in \{100, 400, 800, 1600, 3200, 6400, 12000\}$. Lemmata and inflections are randomly selected from all available paradigms.

**Results and Discussion.** Tab. 3 shows the effectiveness of transfer learning. There are *two* baselines. (i) "0": no transfer, i.e., we consider only in-domain data; (ii) "AR": Arabic, which is unrelated to all target languages.

With the exception of the 200 sample case of ET→SME, cross-lingual transfer is always better than the two baselines; the maximum improvement is 0.58 (0.58 vs. 0.00) in accuracy for the 50 sample case of CA→ES. More closely related source languages improve performance more than distant ones. French, the Romance language least similar to Spanish, performs worst for →ES. For the target language Macedonian, Bulgarian provides most benefit. This can again be explained by similarity: Bulgarian is closer to Macedonian than the other languages in this group. The best result for Ukrainian is RU→UK. Unlike Polish and Slowenian, Russian is the only language in this group that uses the same script as Ukrainian, showing the importance of the alphabet for transfer. Still, the results also demonstrate that transfer works across alphabets (although not as well); this suggests that similar embeddings for similar characters have been learned. Finnish is the language that is closest to Estonian and it again performs best as a source language for Estonian. For Northern Sami, transfer works least well, probably because the distance between sources and target is largest in this case. The distance of the Sami languages from the Finnic (Estonian, Finnish) and Ugric (Hungarian) languages is much larger than the distances within Romance and within Slavic.[4] However, even for Northern Sami, adding an additional language is still always beneficial compared to the monolingual baseline.

Learning curves for Romance and Arabic further support our finding that language similarity is important. In Fig. 2, knowledge is transferred to Spanish, and a baseline – a model trained *only* on Spanish data – shows the accuracy obtained without any transfer learning. Here, Catalan and Italian help the most, followed by Portuguese, French and,

---

[3]For Estonian, we use 7094 (not 12,000) train and 5000 (not 10,000) test samples as more data is unavailable.

[4]We have enlisted the expert for Uralic at our university and are in the process of analyzing SME results in more detail.

| source | Romance | | | | | | Slavic I | | | | | Slavic II | | | | | Uralic I | | | | | Uralic II | | | | |
|---|---|---|---|---|---|---|---|---|---|---|---|---|---|---|---|---|---|---|---|---|---|---|---|---|---|---|
| target | 0 | AR | PT | CA | IT | FR | 0 | AR | RU | BG | CS | 0 | AR | RU | PL | SL | 0 | AR | FI | HU | ET | 0 | AR | FI | HU | SME |
| | →ES | | | | | | →MK | | | | | →UK | | | | | →SME | | | | | →ET | | | | |
| 50 acc | 0.00 | 0.04 | 0.48 | **0.58** | 0.46 | 0.29 | 0.00 | 0.00 | 0.23 | **0.47** | 0.13 | 0.01 | 0.01 | **0.47** | 0.16 | 0.07 | 0.00 | 0.00 | **0.03** | 0.01 | 0.01 | 0.02 | 0.01 | **0.35** | 0.21 | 0.17 |
| 50 ED | 5.42 | 4.06 | 0.85 | **0.80** | 1.15 | 1.82 | 5.71 | 5.59 | 1.61 | **0.87** | 2.32 | 5.23 | 4.80 | **0.77** | 2.14 | 3.12 | 8.47 | 8.64 | **5.41** | 6.53 | 7.03 | 4.50 | 4.51 | **1.55** | 2.19 | 2.60 |
| 200 acc | 0.38 | 0.54 | 0.62 | **0.78** | 0.74 | 0.60 | 0.21 | 0.40 | 0.62 | **0.77** | 0.57 | 0.16 | 0.21 | **0.64** | 0.55 | 0.50 | 0.05 | 0.09 | **0.20** | 0.18 | 0.06 | 0.34 | 0.53 | **0.74** | 0.71 | 0.66 |
| 200 ED | 1.37 | 0.87 | 0.57 | 0.78 | **0.44** | 0.82 | 1.93 | 1.12 | 0.68 | **0.36** | 0.72 | 2.09 | 1.60 | **0.49** | 0.73 | 0.82 | 5.43 | 4.93 | **3.20** | 3.37 | 5.01 | 1.47 | 0.98 | **0.41** | 0.48 | 0.62 |

Table 3: Accuracy (acc) and edit distance (ED) of cross-lingual transfer learning for paradigm completion. The target language is indicated by "→", e.g., it is Spanish for "→ES". Sources are indicated in the row "source"; "0" is the monolingual case. Except for Estonian, we train on $n_s = 12{,}000$ source samples and $n_t \in \{50, 200\}$ target samples (as indicated by the row). There are *two* baselines in the table. (i) "0": no transfer, i.e., we consider only in-domain data; (ii) "AR": the Semitic language Arabic is unrelated to all target languages and functions as a dummy language that is unlikely to provide relevant information. All languages are denoted using the official codes (SME=Northern Sami).

finally, Arabic. This corresponds to the order of lexical similarity with Spanish, except for the performance of Portuguese (cf. Tab. 2). A possible explanation is the potentially confusing overlap of lemmata between the two languages – cf. discussion in the next subsection. That the transfer learning setup improves performance for the unrelated language Arabic as source is at first surprising. But adding new samples to a small training set helps prevent overfitting (e.g., rote memorization) even if the source is a morphologically unrelated language; effectively acting as a regularizer.[5] This will also be discussed below.

**Error Analysis for Romance.** Even for only 50 Spanish instances, many inflections are correctly produced in transfer. For, e.g., (*criar*, 3PlFutSbj) ↦ *criaren*, model outputs are: fr: *criaren*, ca: *criaren*, es: *crntaron*, it: *criaren*, ar: *ecriren*, pt: *criaren* (all correct except for the two baselines). Many errors involve accents, e.g., (*contrastar*, 2PlFutInd) ↦ *contrastaréis*; model outputs are: fr: *contrastareis*, ca: *contrastareis*, es: *conterarían*, it: *contrastareis*, ar: *contastarías*, pt: *contrastareis*. Some inflections all systems get wrong, mainly because of erroneously applying the inflectional rules of the source to the target. Finally, the output of the model trained on Portuguese contains a class of errors that are unlike those of other systems. Example: (*contraatacar*, 1SgCond) ↦ *contraatacaría* with those solutions: fr: *contratacaríam*, ca: *contraatacaría*, es: *concarnar*, it: *contratacé*, ar: *cuntataría* and pt: *contra-atacaría*. The Portuguese model inserts "-" because Portuguese train data contains *contraatacar* and "-" appears in its inflected form.[6]

| | | 0 | PT | CA | IT | FR | AR |
|---|---|---|---|---|---|---|---|
| | | →ES | | | | | |
| one shot | acc | 0.00 | **0.44** | 0.39 | 0.23 | 0.13 | 0.00 |
| one shot | ED | 6.26 | **1.01** | 1.27 | 1.83 | 2.87 | 7.00 |
| zero shot | acc | 0.00 | **0.14** | 0.08 | 0.01 | 0.02 | 0.00 |
| zero shot | ED | 7.18 | **1.95** | 1.99 | 3.12 | 4.27 | 7.50 |

Table 4: Results for one-shot and zero-shot transfer learning. Formatting is the same as for Tab. 3. We still use $n_s = 12000$ source samples. In the one-shot (resp. zero-shot) case, we observe *exactly one form* (resp. *zero forms*) for each tag in the target language at training time.

An example for the generally improved performance across languages for 200 Spanish training samples is (*contrastar*, 2PlIndFut) ↦ *contrastaréis*: all models now produce the correct form.

## 5.2 Exp. 2: Zero-Shot/One-Shot Transfer

In §5.1, we investigated the relationship between in-domain (target) training set size and performance. Here, we look at the extreme case of training set sizes 1 (one-shot) and 0 (zero-shot) for a tag. We train our model on *a single* sample for *half* of the tags appearing in the low-resource language, i.e., if $\mathcal{T}_\ell$ is the set of morphological tags for the target language, train set size is $|\mathcal{T}_\ell|/2$. As before, we add 12,000 source samples.

We report *one-shot accuracy* (resp. *zero-shot accuracy*), i.e., the accuracy for samples with a tag that has been seen once (resp. never) during training. Note that the model has seen the *individual subtags* each tag is composed of.[7]

**Data.** Our experimental setup is similar to §5.1: we use the same dev, test and high-resource train sets as before. However, the low-resource data is created in the way specified above. To remove a potentially confounding variable, we impose the condition that no two training samples belong to

---

[5]Following (Kann and Schütze, 2016b) we did not use standard regularizers. To verify that the effect of Arabic is a regularization effect, we ran a small monolingual experiment on ES (200 setting) with dropout 0.5 (Srivastava et al., 2014). The resulting accuracy is 0.57, very similar to the comparable Arabic number of 0.54 in the table.

[6]To investigate this in more detail we retrain the Portuguese model with 50 Spanish samples, but exclude all lemmata

that appear in Spanish train/dev/test, resulting in only 3695 training samples. Accuracy on test *increases* by 0.09 despite the reduced size of the training set.

[7]It is very unlikely that due to random selection a subtag will not be in train; this case did not occur in our experiments.

the same lemma.

**Results and Discussion.** Tab. 4 shows that the Spanish and Arabic systems do not learn anything useful for either half of the tags. This is not surprising as there is not enough Spanish data for the system to generalize well and Arabic does not contribute exploitable information. The systems trained on French and Italian, in contrast, get a non-zero accuracy for the zero-shot case as well as 0.13 and 0.23, respectively, in the one-shot case. This shows that a single training example is sometimes sufficient for successful generation although generalization to tags never observed is rarely possible. Catalan and Portuguese show the best performance in both settings; this is intuitive since they are the languages closest to the target (cf. Tab. 2). In fact, adding Portuguese to the training data yields an absolute increase in accuracy of 0.44 (0.44 vs. 0.00) for one-shot and 0.14 (0.14 vs. 0.00) for zero-shot with corresponding improvements in edit distance.

Overall, this experiment shows that with transfer learning from a closely related language the performance of zero-shot morphological generation improves over the monolingual approach, and, in the one-shot setting, it is possible to generate the right form nearly half the time.

### 5.3 Exp. 3: True Transfer vs. Other Effects

We would like to separate the effects of regularization that we saw for Arabic from true transfer.

To this end, we generate a random cipher (i.e., a function $\gamma : \Sigma \cup \mathcal{S} \mapsto \Sigma \cup \mathcal{S}$) and apply it to all word forms and morphological tags of the high-resource train set; target language data are not changed. Ciphering makes it harder to learn true "linguistic" transfer of morphology. Consider the simplest case of transfer: an identical mapping in two languages, e.g., (*visitar*, 1SgPresInd) $\mapsto$ *visito* in both Portuguese and Spanish. If we transform Portuguese using the cipher $\gamma(\text{iostv}...) = \text{kltqa}...$, then *visito* becomes *aktkql* in Portuguese and its tag becomes similarly unrecognizable as being identical to the Spanish tag 1SgPresInd. Our intuition is that ciphering will disrupt transfer of morphology.[8] On the other hand, the regularization effect we observed with Arabic should still be effective.

**Data.** We use the Portuguese-Spanish and

---

[8]Note that ciphered input is much harder than transfer between two alphabets (Latin/Cyrillic) because it creates ambiguous input. In the example, Spanish "i" is totally different from Portuguese "i" (which is really "k"), but the model must use the same representation.

Arabic-Spanish data from Experiment 1. We generate a random cipher and apply it to morphological tags and word forms for Portuguese and Arabic. The language tags are kept unchanged. Spanish is also not changed. For comparability with Tab. 3, we use the same dev and test sets as before.

**Results and Discussion.** Tab. 5 shows that performance of PT→ES drops a lot: from 0.48 to 0.09 for 50 samples and from 0.62 to 0.54 for 200 samples. This is because there are no overt similarities between the two languages left after applying the cipher, e.g., the two previously identical forms *visito* are now different.

The impact of ciphering on AR→ES varies: slightly improved in one case (0.54 vs. 0.56), slightly worse in three cases. We also apply the cipher to the tags and Arabic and Spanish share subtags, e.g., Sg. Just the knowledge that something is a subtag is helpful because subtags must not be generated as part of the output. We can explain the tendency of ciphering to decrease performance on AR→ES by the "masking" of common subtags.

For 200 samples and ciphering, there is no clear difference in performance between Portuguese and Arabic. However, for 50 samples and ciphering, Portuguese (0.09) seems to perform better than Arabic (0.02) in accuracy. Portuguese uses suffixation for inflection whereas Arabic is templatic and inflectional changes are not limited to the end of the word. This difference is not affected by ciphering. Perhaps even ciphered Portugese lets the model learn better that the beginnings of words just need to be copied. For 200 samples, the Spanish dataset may be large enough, so that ciphered Portuguese no longer helps in this regard.

Comparing no transfer with transfer from a ciphered language to Spanish, we see large performance gains, at least for the 200 sample case: 0.38 (0→ES) vs. 0.54 (PT→ES) and 0.56 (AR→ES). This is evidence that our conjecture is correct that the baseline Arabic mainly acts as a regularizer that prevents the model from memorizing the training set and therefore improves performance. So performance improves even though no true transfer of morphological knowledge takes place.

## 6 Related Work

**Cross-lingual transfer learning** has been used for many tasks: automatic speech recognition (Huang et al., 2013), parsing (Cohen et al., 2011; Søgaard, 2011; Naseem et al., 2012), entity recog-

|  |  | 0→ES | PT→ES | | AR→ES | |
|---|---|---|---|---|---|---|
|  |  |  | orig | ciph | orig | ciph |
| 50 | acc | 0.00 | 0.48 | 0.09 | 0.04 | 0.02 |
|  | ED | 5.42 | 0.85 | 3.25 | 4.06 | 4.62 |
| 200 | acc | 0.38 | 0.62 | 0.54 | 0.54 | 0.56 |
|  | ED | 1.37 | 0.57 | 0.95 | 0.87 | 0.93 |

Table 5: Results for ciphering. "0→ES" and "orig" are original results, copied from Tab. 3; "ciph" is the result after the cipher has been applied.

nition (Wang and Manning, 2014b) and machine translation (Johnson et al., 2016; Ha et al., 2016). One straightforward method is to translate datasets and then train a monolingual model (Fortuna and Shawe-Taylor, 2005; Olsson et al., 2005). Also, aligned corpora have been used to project information from annotations in one language to another (Yarowsky et al., 2001; Padó and Lapata, 2005). The drawback is that machine translation errors cause errors in the target. Therefore, alternative methods have been proposed, e.g., to port a model trained on the source language to the target language (Shi et al., 2010).

In the realm of morphology, Buys and Botha (2016) recently adapted methods for the training of POS taggers to learn weakly supervised morphological taggers with the help of parallel text. Snyder and Barzilay (2008a, 2008b) developed a non-parametric Bayesian model for morphological segmentation. They performed identification of cross-lingual abstract morphemes and segmentation simultaneously and reported, similar to us, best results for related languages.

Work on **paradigm completion** has recently been encouraged by the SIGMORPHON 2016 shared task on morphological reinflection (Cotterell et al., 2016a). Some work first applies an unsupervised alignment model to source and target string pairs and then learns a string-to-string mapping (Durrett and DeNero, 2013; Nicolai et al., 2015), using, e.g., a semi-Markov conditional random field (Sarawagi and Cohen, 2004). Encoder-decoder RNNs (Aharoni et al., 2016; Faruqui et al., 2015; Kann and Schütze, 2016b), a method which our work further develops for the cross-lingual scenario, define the current state of the art.

**Encoder-decoder RNNs** were developed in parallel by Cho et al. (2014) and Sutskever et al. (2014) for machine translation and extended by Bahdanau et al. (2015) with an attention mechanism, supporting better generalization. They have been applied to NLP tasks like speech recognition (Graves and Schmidhuber, 2005; Graves et al., 2013), parsing (Vinyals et al., 2015) and segmentation (Kann et al., 2016). More recently, a number of papers have used encoder-decoder RNNs in *multitask and transfer learning settings*; this is mainly work in machine translation (MT): (Dong et al., 2015; Zoph and Knight, 2016; Chu et al., 2017; Johnson et al., 2016; Luong et al., 2016; Firat et al., 2016; Ha et al., 2016), *inter alia*. Each of these papers has both similarities and differences with our approach. (i) Most train several distinct models whereas we train a *single model* on input augmented with an explicit encoding of the language (similar to (Johnson et al., 2016)). (ii) Let $k$ and $m$ be the number of different input and output languages. We address the case $k \in \{1, 2\}$ and $m = k$. Other work has addressed cases with $k > 2$ or $m > 2$; this would be an interesting avenue of future research for paradigm completion. (iii) Whereas training RNNs in MT is hard, we only experienced one difficult issue in our experiments (due to the low-resource setting): regularization. (iv) Some work is word- or subword-based, our work is character-based. The same way that similar word embeddings are learned for the inputs *cow* and *vache* (French for "cow") in MT, we expect similar embeddings to be learned for similar Cyrillic/Latin characters. (v) Similar to work in MT, we show that zero-shot (and, by extension, one-shot) learning is possible.

(Ha et al., 2016) (which was developed in parallel to our transfer model although we did not prepublish our paper on arxiv) is most similar to our work. Whereas Ha et al. (2016) address MT, we focus on the task of paradigm completion in low-resource settings and establish the state of the art for this problem.

# 7 Conclusion

We presented a cross-lingual transfer learning method for paradigm completion, based on an RNN encoder-decoder model. Our experiments showed that information from a high-resource language can be leveraged for paradigm completion in a related low-resource language. Our analysis showed that the degree to which the source language data helps for a certain target language depends on their relatedness. Our method led to significant improvements in settings with limited training data – up to 58% absolute improvement in accuracy – and, thus, enables the use of state-of-the-art models for paradigm completion in low-resource languages.

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
