# Peer review of "One-Shot Neural Cross-Lingual Transfer for Paradigm Completion"

_ACL 2017 — decision unknown_

[Official Review · Reviewer 1 · rating 4 · confidence 4]
soundness 5 · originality 5 · clarity 5 · impact 3 · substance 4 · appropriateness 5 · meaningful comparison 3 · presentation format Poster

The paper introduces a simple and effective method for morphological paradigm
completion in low-resource settings. The method uses a character-based seq2seq
model trained on a mix of examples in two languages: a resource-poor language
and a closely-related resource-rich language; each training example is
annotated with a paradigm properties and a language ID. Thus, the model enables
transfer learning across languages when the two languages share common
characters and common paradigms. While the proposed multi-lingual solution is
not novel (similar architectures have been explored in syntax, language
modeling, and MT), the novelty of this paper is to apply the approach to
morphology. Experimental results show substantial improvements over monolingual
baselines, and include a very thorough analysis of the impact of language
similarities on the quality of results. The paper is interesting, very clearly
written, I think it’ll be a nice contribution to the conference program. 

Detailed comments: 

— My main question is why the proposed general multilingual methodology was
limited to pairs of languages, rather than to sets of similar languages? For
example, all Romance languages could be included in the training to improve
Spanish paradigm completion, and all Slavic languages with Cyrillic script
could be mixed to improve Ukrainian. It would be interesting to see the
extension of the models from bi-lingual to multilingual settings. 

— I think Arabic is not a fair (and fairly meaningless) baseline, given how
different is its script and morphology from the target languages. A more
interesting baseline would be, e.g., a language with a partially shared
alphabet but a different typology. For example, a Slavic language with Latin
script could be used as a baseline language for Romance languages. If Arabic is
excluded, and if we consider a most distant language in the same the same
family as a baseline, experimental results are still strong. 

— A half-page discussion of contribution of Arabic as a regularizer also adds
little to the paper; I’d just remove Arabic from all the experiments and
would add a regularizer (which, according to footnote 5, works even better than
adding Arabic as a transfer language).              

— Related work is missing a line of work on “language-universal” RNN
models that use basically the same approach: they learn shared parameters for
inputs in multiple languages, and add a language tag to the input to mediate
between languages. Related studies include a multilingual parser (Ammar et al.,
2016), language models (Tsvetkov et al., 2016), and machine translation
(Johnson et al., 2016 )

Minor: 
— I don’t think that the claim is correct in line 144 that POS tags are
easy to transfer across languages. Transfer of POS annotations is also a
challenging task.  

References: 

Waleed              Ammar, George Mulcaire, Miguel Ballesteros, Chris Dyer, and Noah
A.
Smith. "Many languages, one parser.” TACL 2016. 

Yulia Tsvetkov, Sunayana Sitaram, Manaal Faruqui, Guillaume Lample, Patrick
Littell, David Mortensen, Alan W. Black, Lori Levin, and Chris Dyer. "Polyglot
neural language models: A case study in cross-lingual phonetic representation
learning.” NAACL 2016.

Melvin Johnson, Mike Schuster, Quoc V. Le, Maxim Krikun, Yonghui Wu, Zhifeng
Chen, Nikhil Thorat et al. "Google's Multilingual Neural Machine Translation
System: Enabling Zero-Shot Translation." arXiv preprint arXiv:1611.04558 2016.

-- Response to author response: 

Thanks for your response & I'm looking forward to reading the final version!